# The Strategic Use of an Immunomodulatory Feed Additive in Supplements for Grazing Young Nellore Bulls Transported after Weaning: Performance, Physiological, and Stress Parameters

Luis Henrique Curcino Batista [1,*], Ivanna Morais Oliveira [2], Laura Franco Prados [2], Laylles Costa Araújo [1], Igor Machado Ferreira [1], Mateus José Inácio de Abreu [1], Saulo Teixeira Rodrigues de Almeida [1], César Aparecido de Araújo Borges [3], Gustavo Rezende Siqueira [1,2] and Flávio Dutra de Resende [1,2]

[1] Department of Animal Science, Faculty of Agricultural and Veterinary Sciences, São Paulo State University (UNESP), Jaboticabal 14884-900, São Paulo, Brazil; laylles_araujo@hotmail.com (L.C.A.); igor.machado@unesp.br (I.M.F.); abreu.mateusji@gmail.com (M.J.I.d.A.); saulo.almeida@unesp.br (S.T.R.d.A.); siqueiragr@sp.gov.br (G.R.S.); fresende@sp.gov.br (F.D.d.R.)
[2] Agência Paulista de Tecnologia dos Agronegócios (APTA), Colina 14770-000, São Paulo, Brazil; imoraesdeoliveira@yahoo.com.br (I.M.O.); laurafrancoprados@hotmail.com (L.F.P.)
[3] Phibro Animal Health, Campinas 13025-170, São Paulo, Brazil; cesar.borges@realbeef.com.br
[*] Correspondence: luishenrique_94cb@hotmail.com

**Abstract:** The objective of this study was to evaluate four different feeding strategies using an immunomodulatory feed additive for newly weaned Nellore cattle, before and after road transport, on their physiological parameters and performances during the growing phase of pastures. In total, eighty-four young Nellore bulls (initial BW = 174 ± 11 kg; 7 ± 1 months of age) were blocked by their initial body weights 42 days before road transport (d −42) and randomly assigned to one of the four supplementation strategies. The treatments were: (1) Control (CON): no immunomodulatory feed additive (NUTRA) supplementation; (2) NUTRA pre: the inclusion of NUTRA only in the pre-transport period (d −42 to d 0); (3) NUTRA post: the inclusion of NUTRA for 42 days, only in the post-transport period (d 0 to d 42); and (4) NUTRA growth: the inclusion of NUTRA during the whole experimental period (d −42 to d 210). On d 0, the calves were transported on dirty roads in a commercial livestock trailer for 200 km (8 h). There was no effect of the treatments on the animal performance or the physiological parameters in their plasma. However, there were effects on the day of the blood sampling for all the parameters. The highest concentration of cortisol was observed on d 3 post-transport (129 ng/mL) and this decreased over time (22.4 ng/mL; d 210). On the other hand, their glucose peaked at unloading, with lower concentrations on d 7 and d 14. Their total protein concentrations increased from d 0 to d 7. The immunomodulatory feed additive supplementation at 10 g/100 kg BW/day did not modulate the physiological responses in their plasma and did not influence the performance of the Nellore bulls during the growing phase of their pastures.

**Keywords:** *Bos indicus*; NutraGen; stress; supplementation; weaned calves

## 1. Introduction

Long-distance transport is frequent and often inevitable within the beef industry. This is mainly due to the geographical separation of cow–calf regions, growth phases, and finishing operations [1,2]. One of the most widely recognized stressors in beef cattle production is transport [1,3]. In weaned calves, it has been shown that transport is perceived as an acute stress that increases their serum cortisol concentrations [4] and alters their energy and protein metabolism [5]. Therefore, transport may change and impair animal growth rate, which leads to a greater susceptibility to diseases [6,7]. Thus, nutritional strategies that prevent these stress-related physiological disorders caused during transport may be advantageous to the well-being, health, and productivity of beef cattle [7].

Feed additives with immunomodulatory properties have been used to improve the health and productivity parameters in dairy cattle and sheep during stressful periods [8–10]. Lippolis et al. [11] reported a lower plasma cortisol concentration and enhanced innate immunity in newly weaned feedlot steers that were supplemented with an immunomodulatory feed additive after road transport. Under the conditions of thermal stress, dairy cows supplemented with an immunomodulatory feed additive showed positive changes in their energy, protein, and mineral metabolism, in addition to an increased forage intake, milk yield, and milk fat [12]. Moreover, Wu et al. [13] reported a reduced incidence of mastitis, a lower body weight loss during early lactation, and a lower somatic cell count in the milk of transition cows that were supplemented with an immunomodulatory feed additive compared to non-supplemented animals.

Previous studies have suggested the potential benefits of immunomodulatory feed additive supplementation for animals under stressful conditions. However, to our knowledge, studies evaluating the effects of this immunomodulatory feed additive supplementation for grazing calves transported after weaning, especially for Nellore cattle, are scarce. A supplementation with the feed additive evaluated in this study, several weeks before the stress or immune challenge, has been recommended [14]. Therefore, more research is needed to evaluate the benefits of an immunomodulatory feed additive supplementation for weaned Nellore cattle under grazing conditions.

We hypothesized that an immunomodulatory feed additive supplementation during pre- and post-transport would decrease the plasma cortisol concentrations of the cattle, changing their metabolism and improving the animal performance during the growing phase. The objective of this experiment was to evaluate different immunomodulatory feed additive supplementation strategies, pre- and post-transport, for weaned Nellore calves that were maintained on a pasture, and their effects on the physiological parameters and performances during the growing phase.

## 2. Materials and Methods

### 2.1. Local and Climate

The experiment was conducted at the Agência Paulista de Tecnologia dos Agronegócios (APTA), Colina, SP, Brazil (20°43′5″ S, 48°32′38″ W). During the pre-transport period, the temperature ranged from 11.8 °C to 29.5 °C and the total rainfall was 38.8 mm, distributed over seven days. During the post-transport period, the temperature ranged from 16.9 °C to 32.4 °C and the total rainfall was 984 mm, distributed over 96 days.

### 2.2. Experimental Period

The animals were evaluated from August 2018 to April 2019. The experimental period was divided into pre- and post-transport during the growing phase. The pre-transport period lasted 42 days (beginning on d −42; Figure 1) and the post-transport period lasted 210 days (beginning on d 0), which was divided into five periods of 42 days each (d 0, d 42, d 84, d 126, d 168 and d 210; Figure 1).

### 2.3. Experimental Area

The experimental area was composed of *Urochloa brizantha* cv. Marandu and was divided into 12 paddocks (2.2–2.4 ha each). All the paddocks were equipped with water tanks (1000 L) and feeders (27 linear cm/animal) for the supplementation. A urea-based fertilizer was applied at a rate of 90 kg/ha of nitrogen during the experimental period and was divided into two applications of 45 kg/ha in February and March 2019.

### 2.4. Animals

In total, eighty-four non-castrated Nellore calves (weaning BW = 184 ± 19 kg; 7 ± 1 months of age) were used in a randomized complete block design (blocked by initial BW). The animals belonged to the same herd and were managed similarly until the beginning of the experiment. The calves were weaned two weeks before the beginning

of the experiment and were allowed to keep fence line contact with the dams for seven days, according to the recommendations by Taylor et al. [15]. This interval was considered to be a transition period between the weaning and experimental procedure to reduce any changes in the animal behavior due to weaning [16].

Prior to the experimental period, all the animals were vaccinated against clostridial diseases: *Clostridium chauvoei*, *Clostridium botulinum* types C and D, *Clostridium septicum*, *C. novyi*, *C. perfringens* types B, C, and D, and *C. sordelli* (Poli-Star®; Vallee SA, Minas Gerais, Brazil). They were also administered an anthelmintic: Eprinomectina 5% (LongRange®; Boehringer Ingelheim, São Paulo, Brazil) at 1 mL/50 kg BW, and they were treated against ectoparasites using Fipronil (Topline® Pour-on; Boehringer Ingelheim, São Paulo, Brazil) at a concentration of 1 g/100 mL. The same protocol was repeated on d 42, along with a vaccination against foot-and-mouth disease (Ourovac® Aftosa; Ourofino Saúde Animal Ltda., Cravinhos, São Paulo, Brazil).

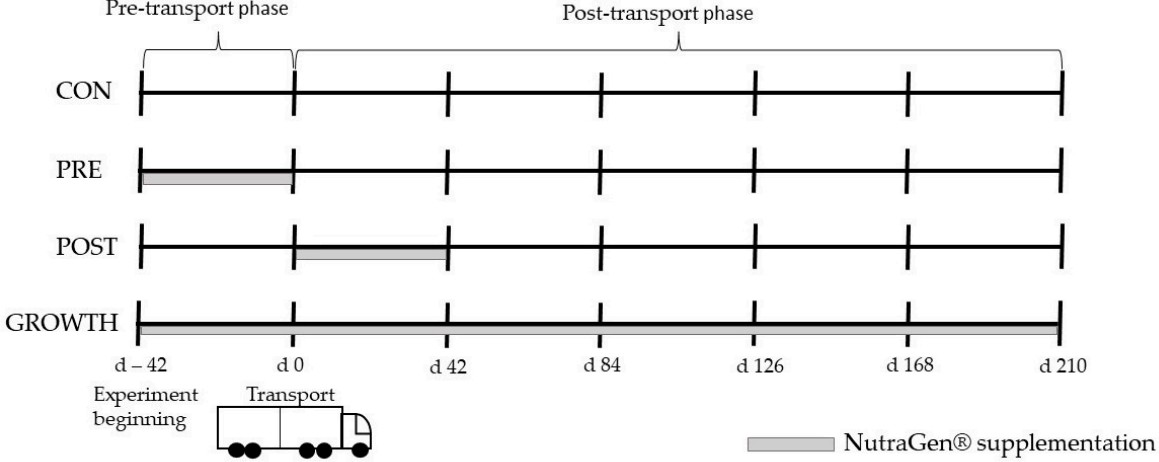

**Figure 1.** Schematic representation of experimental period and different strategies of inclusion of an immunomodulatory feed additive (NutraGen®; Phibro Animal Health, Guarulhos, São Paulo, Brazil) added to supplements at 10 g/100 kg BW/day for Nellore calves submitted to post-weaning transport.

## 2.5. Experimental Design and Treatments

A total of four treatments were evaluated in a randomized complete block design. The animals were randomly distributed into 4 paddocks per block and 7 animals per paddock. The blocks were established based on their BWs (high, medium, and low) and the paddock was considered to be the experimental unit. The treatments were based on different supplementation strategies with an immunomodulatory feed additive (NutraGen® (NUTRA); Phibro Animal Health, Guarulhos, São Paulo, Brasil) during the pre- and post-transport periods in the growth phase (Figure 1): (1) Control (CON): no immunomodulatory supplementation during the entire experiment; (2) NUTRA pre: the inclusion of the immunomodulatory feed additive for 42 days, only in the pre-transport period (d −42 to d 0); (3) NUTRA post: the inclusion of the immunomodulatory feed additive for 42 days, only in the post-transport period (d 0 to d 42); and (4) NUTRA growth: the inclusion of the immunomodulatory feed additive during the entire experiment.

## 2.6. Supplementation and Immunomodulatory Feed Additive

From d −42 to d 42, all the animals received a dry season protein energy supplement (crude protein = 300 g/kg; total digestible nutrients = 650 g/kg), and from d 43 to d 210, the animals received a rainy season protein energy supplement (crude protein = 200 g/kg and total digestible nutrients = 650 g/kg). The supplementation was provided daily at 09:00 at 3 g/kg of the BWs (Table 1).

**Table 1.** Ingredient and nutrient composition of supplements.

| Item | Protein-Energy Supplement [1] | |
|---|---|---|
| | (d −42 to d 42) | (d 43 to d 210) |
| Ingredients (g/kg) | | |
| Corn | 167 | 484 |
| Sorghum | 50.0 | 147 |
| Soybean meal | 429 | 125 |
| Corn gluten meal (21% CP) | 150 | - |
| Rice bran | 100 | 100 |
| Slow-release urea | 10.0 | 5.0 |
| Urea | 4.0 | 21.0 |
| Sodium chloride | 35.0 | 36.0 |
| Mineral mixture | 48.7 | 75.8 |
| Vmax2® [2] | 6.6 | 6.6 |
| Composition as feed (g/kg) | | |
| Dry matter | 89.0 | 89.6 |
| Crude protein (minimum) | 30.0 | 20.0 |
| Protein equivalent from NPN | 12.5 | 8.13 |
| Total digestible nutrients (minimum) | 65.0 | 65.0 |

NPN = non-protein nitrogen. [1] Provided daily at 3 g/kg of body weight per animal. In both supplements, the macro and micro mineral guaranteed levels were: calcium (minimum) 15 g/kg, calcium (maximum) 25 g/kg, cobalt (minimum) 8.7 mg/kg, copper (minimum) 100 mg/kg, sulfur (minimum) 5000 mg/kg, fluorine (maximum) 70 mg/kg, phosphorus (minimum) 7000 mg/kg, iodine (minimum) 5.30 mg/kg, magnesium (minimum) 1000 mg/kg, and manganese (minimum) 68 mg/kg. [2] Guaranteed analysis of Vmax2® Phibro = 20 g of virginiamycin per kg of product.

The immunomodulatory feed additive was added to the supplements at 10 g/100 kg of the BWs. The inclusion level of NUTRA was based on previous studies [10,17]. The additive contained a mixture of active dry *Saccharomyces cerevisiae*, dried *Trichoderma longibrachiatum* fermentation extract, niacin supplement, vitamin B12, riboflavin-5-phosphate, D-calcium pantothenate, choline chloride, biotin, thiamine monohydrate, pyridoxine hydrochloride, menadione dimethylpyrimidinol bisulfite, folic acid, calcium aluminum silicate, sodium aluminum silicate, diatomaceous earth, calcium carbonate, rice hulls, and mineral oil. The additive was mixed with the supplement every three days in a horizontal ribbon mixer for 5 min.

*2.7. Road Transport*

On d 0, all the animals were transported for 8 h over 200 km of unpaved roads. The transport was conducted over three different days to allow for the sampling (observing the supplementation period of 42 days pre-transport). Animals of the same blocks were transported in the same commercial livestock trailers, with dimensions of 7.40 m long × 2.40 m wide × 1.90 m high, allowing for a density of 1.6 animals/m$^2$ and 339, 296, and 270 kg of BW/m$^2$, respectively, for the high-, medium-, and low-BW animals, who were weighed before transport.

At around 06:00, the animals were removed from the paddocks, mixed, taken to the corral, and weighed. At 07:00, the animals were loaded and transported throughout the day. There was a one-hour break at around 12:00. At the end of the day, the animals were unloaded, weighed, and their blood samples were taken. After the sampling, the animals were placed into the same paddocks, maintaining the same groups of animals prior to the transport.

### 2.8. Grazing Method

A continuous grazing system with a variable stocking rate was used to maintain the same forage sward height for all the treatments (around 25 cm) [18]. This sward height corresponded to the 95% light interception for *Urochloa brizantha* cv. Marandu, in which the productivity parameters were highest [19]. Put-and-take steers were used to control the sward height [18] after the post-transport period, when the pasture parameters were characteristic of the rainy season, i.e., there were higher rates of green leaf accumulation [20]. These animals came from the same herd as the experimental animals and were kept in an adjacent area with access to the supplementation throughout the experiment.

### 2.9. Quantitative and Qualitative Evaluation of the Pasture

The forage mass was estimated at the beginning of the experiment (d −42) and every 42 days (d 0, d 42, d 84, d 126, d 168, and d 210), using the double sampling method [21]. The quantitative and structural components of the forage sward were evaluated at medium heights. The forage samples were separated into green leaf, dead leaf, green stem, and dead stem in the pre-transport period (dry season), and green leaf, green stem, and dead material in the post-transport period (rainy season). After the separation, the forage components were weighed and oven-dried at 55 °C for 72 h to obtain the partial DM and the proportion of each component in the forage sward (Table 2).

**Table 2.** Characteristics of the forage sward grazed by young Nellore bulls during growth phase, subjected to different immunomodulatory feed supplementation strategies in the pre-and post-transport periods.

| Item | Pre-Transport | | | | | Post-Transport | | | | |
|---|---|---|---|---|---|---|---|---|---|---|
| | Treatments [1] | | | | SEM | Treatments [1] | | | | SEM |
| | CON | Pre | Post | Growth | | CON | Pre | Post | Growth | |
| Quantitative characteristics (n = 12) | | | | | | | | | | |
| Sward height (cm) | 21.3 | 21.4 | 16.4 | 18 | 2.74 | 25.2 | 24.7 | 22.7 | 23.1 | 1.85 |
| Forage mass (kg DM/ha) | 4888 | 4356 | 3509 | 3585 | 705 | 4030 | 3767 | 3526 | 3502 | 337 |
| Green leaf (% DM) | 8.66 | 11.5 | 7.53 | 13.1 | 3.2 | 42.1 | 45.3 | 48 | 48.8 | 3.88 |
| Green stem (% DM) | 9.85 | 8.33 | 5.95 | 6.56 | 1.49 | 22.4 | 21.2 | 21.2 | 23.9 | 1.21 |
| Dead leaf (% DM) | 33.5 | 28.3 | 33 | 28.1 | 2.33 | - | - | - | - | - |
| Dead stem (% DM) | 48 | 51.9 | 53.5 | 52.3 | 3.95 | - | - | - | - | - |
| Dead material (% DM) | - | - | - | - | - | 35.5 | 33.5 | 30.7 | 27.3 | 4.66 |
| Forage allowance (kg DM/kg BW) | 8.15 | 7.3 | 5.6 | 5.86 | 1.14 | 4.11 | 4.02 | 3.79 | 3.67 | 0.35 |
| Forage allowance kg (GLDM/kg BW) | 0.29 | 0.25 | 0.17 | 0.24 | 0.06 | 1.7 | 1.82 | 1.86 | 1.76 | 0.16 |
| Stocking rate (AU/ha) | 1.22 | 1.22 | 1.2 | 1.19 | 0.06 | 2.14 | 2.01 | 1.88 | 1.97 | 0.1 |
| Qualitative characteristics (g/kg) (n = 12) | | | | | | | | | | |
| Crude protein | 70.8 | 65.9 | 73.7 | 72.1 | 4.80 | 123 | 110 | 120 | 117 | 5.60 |
| NDF | 699 | 702 | 689 | 703 | 10.6 | 623 | 632 | 617 | 620 | 5.50 |
| ADF | 333 | 331 | 334 | 338 | 5.40 | 274 | 274 | 271 | 264 | 5.50 |
| Lignin | 42.3 | 41.9 | 42.7 | 44.9 | 1.90 | 30.7 | 30.6 | 28.7 | 29.2 | 1.00 |
| IVDMD | 676 | 677 | 684 | 670 | 9.70 | 807 | 804 | 812 | 816 | 5.60 |

GLDM = green leaf dry matter; IVDMD = in vitro dry matter digestibility. [1] CON: no immunomodulatory supplement; NUTRA pre: inclusion of NutraGen® (10 g/100 kg BW/day) only in the pre-transport period for 42 days; NUTRA post: inclusion of NUTRA (10 g/100 kg BW/day) only in the post-transport period for 42 days; and NUTRA growth: inclusion of NUTRA (10 g/100 kg BW/day) during the whole experimental period (pre-and post-transport).

Every 42 days, hand-plucked samples were used to estimate the dietary nutritional value [22]. The samples were oven-dried at 55 °C for 72 h and then ground in a Wiley mill (Thomas Model 4, Thomas Scientific, Swedesboro, NJ, USA) to pass through a 1 mm mesh sieve. These samples were used for the determination of the CP levels (method 978.04; AOAC, 1995). The NDF and ADF contents were evaluated by a sequential analysis, as described by Van Soest et al. [23]. The cellulose was solubilized using 72% sulfuric acid, whereby the lignin content was obtained by the difference from the ADF. The true in vitro DM digestibility was determined, according to Van Soest and Robertson [24].

The quantitative and qualitative characteristics of the forage were subjected to statistical analyses, and there was no difference between the analyzed treatments ($p > 0.10$), with the averages being shown in Table 2.

### 2.10. Animal Performance

The animals were weighed after fasting for 16 h on d −42, d 42, d 84, d 126, d 168, and d 210 for the determination of their BWs and average daily gains (ADG). On d 0, the animals were weighed immediately before and after transport to evaluate if their BWs shrunk during transport (%). The shrunk BW from each period was also used to adjust the supplement amounts.

### 2.11. Blood Samples and Analyzes

The blood samples were collected via a jugular venipuncture into tubes (Vacuplast, 9 mL; Weihai Hongyu Medical Devices Co., Ltd., Weihai, China) with spray-dried sodium heparin. The samplings were performed on all the animals on d −42 and d −1 (pretransport), d 0 (unloading), and d 3, d 7, d 14, d 42, and d 210 (post-transport).

The blood samples on d −42, d −1, d 3, d 7, d 14, d 42, and d 210 were taken at 08:00, before the supplementation. These samples were taken to evaluate the period in which the cattle were transported and arrived in a new place, facing stress-induced metabolic changes [7]. The blood samples were stored in ice boxes immediately after their collection and then centrifuged at $2500 \times g$ at 4 °C for 15 min. The blood samples were processed immediately after the blood collection and aliquots of the plasma were stored at −80 °C until further analysis.

All the samples were analyzed for cortisol by a radioimmunoassay (RIA) using a solid-phase commercial kit (Coat-a-count®, Diagnostic Products Corporation, Los Angeles, CA, USA); the urea, albumin, and total proteins were analyzed using commercial biochemical kits (Bioclin; Quibasa-Química Básica Ltda., Belo Horizonte, MG, Brazil) and a biochemical analyzer (Cobas Mira Plus; Roche Diagnostic Systems, Montreuil, France). The plasma samples (d −42, d −1, d 3, d 7, d 14, and d 210) were analyzed for glucose and aspartate aminotransferase (AST) (Bioclin; Quibasa-Química Basica Ltda., Belo Horizonte, MG, Brazil). The plasma samples (d −42, d −1, d 0, and d 210) were analyzed for creatinine, cholesterol, calcium, phosphorus, and magnesium (Bioclin; Quibasa-Química Basica Ltda., Belo Horizonte, MG, Brazil). The intra-and inter-assay variations of the plasma cortisol were 4.31% and 5.2%, respectively.

### 2.12. Statistical Analysis

All the data were analyzed as randomized complete block designs, using the MIXED procedures of SAS® University Edition software (SAS Institute, Cary, North Carolina, US). Each paddock was considered to be an experimental unit, where the bull (paddock) and paddock (treatment) were included as random effects in all the analyses. The variables, when evaluated over the experimental periods (ADG and blood parameters), were analyzed as repeated measures and tested for the fixed effects of treatment, time, and resulting interactions, using the paddock (treatment) as the subject. Different covariance structures were tested with the final choice, depending on the lowest value for the Akaike information criterion. The variables that were not evaluated by period were: the initial BW, final BW, and BW shrink (%), and were used in the model as fixed effects only for the effects of the treatments. All the results are reported as least squares means. The data were separated using PDIFF if a significant F-test was detected. The significance was set at $p \leq 0.05$, and tendencies were noted if $p > 0.05$ and $\leq 0.10$.

## 3. Results

There was no interaction between the treatment and period ($p \geq 0.86$) in the post-transport ADG (d 0 to d 210) and total ADG (d −42 to d 210). There was no effect of the treatments ($p \geq 0.22$) on the ADGs and BWs during the experimental period (Table 3). There was no difference between the treatments ($p = 0.81$) for the BW shrink (Table 3). There was an effect ($p < 0.01$) of the periods on the ADG (Table 3). The ADG was lower in the pre-transport (mean of 0.100 kg/d) than in the post-transport period. The highest ADG was observed between d 168 and d 210 (mean of 1.08 kg/d).

**Table 3.** Performance of young Nellore bulls subjected to different immunomodulatory feed supplementation strategies in the pre- and post-transport periods during the growth phase.

| Item | Treatments [1] | | | | SEM | *p*-Value |
|---|---|---|---|---|---|---|
| | **CON** | **Pre** | **Post** | **Growth** | | |
| Initial BW (d −42) (kg) | 173 | 174 | 174 | 174 | 11.1 | 0.45 |
| Post-transport BW (d 0; kg) | 181 | 178 | 179 | 177 | 12.4 | 0.61 |
| BW shrink (%) | 6.98 | 7.32 | 6.54 | 7.04 | 0.61 | 0.81 |
| Final BW (d 210; kg) | 359 | 356 | 359 | 356 | 15.1 | 0.85 |
| Pre-transport ADG (kg/day) | 0.15 | 0.09 | 0.11 | 0.07 | 0.03 | 0.22 |
| Post-transport ADG (kg/day) | 0.87 | 0.87 | 0.88 | 0.87 | 0.03 | 0.96 |
| Total ADG (d −42–d 210; kg/day) | 0.75 | 0.74 | 0.76 | 0.74 | 0.02 | 0.93 |

ADG = average daily gain. [1] CON: no immunomodulatory supplementation; NUTRA pre: inclusion of NU-TRA (10 g/100 kg BW/day) only in the pre-transport period for 42 days; NUTRA post: inclusion of NUTRA (10 g/100 kg BW/day) only in the post-transport period for 42 days; and NUTRA growth: inclusion of NUTRA (10 g/100 kg BW/day) during the whole experimental period (pre- and post-transport). Note: There was an effect of period ($p < 0.01$). There was no significant interaction between treatment and period for post-transport ADG and total ADG ($p \geq 0.86$).

There was a significant effect of the periods ($p < 0.01$) on the BWs (Figure 2). However, the initial BWs (d −42; 174 kg) and BWs on d 0 (179 kg) did not differ ($p = 0.16$) among the treatments. As expected, the BWs were higher ($p < 0.01$) in the following periods due to the greater ADGs.

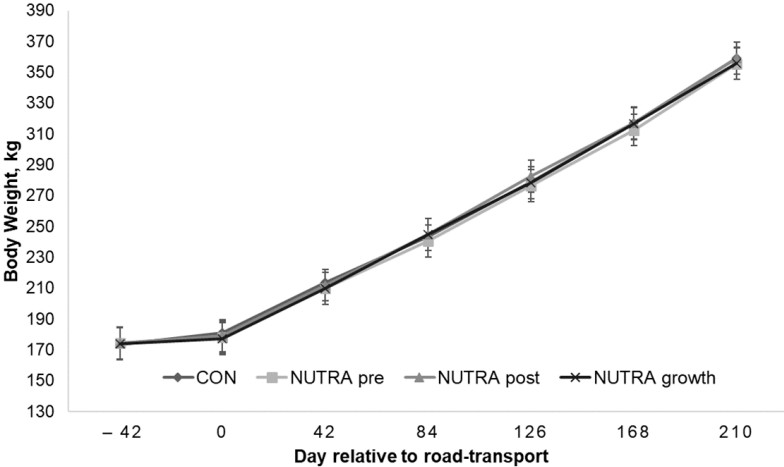

**Figure 2.** Body weight (BW) of young Nellore bulls subjected to different immunomodulator supplementation strategies in the pre- and post-transport during the growth phase. Additive: NutraGen added to the supplement at 10 g/100 kg BW/d. Treatments: ($p = 0.70$); period: $p < 0.01$; and treatment × period ($p = 0.93$). Animals were weighed before and after a 16 h shrink, except on d 0, in which BW was obtained after transport for 8 h.

There was no interaction between the treatment and period ($p \geq 0.17$), nor was there a treatment effect ($p \geq 0.14$) on any of the physiological variables that were measured in the plasma (Table 4). The sampling period influenced ($p < 0.01$) all the physiological variables in the plasma (Figures 3 and 4). The cortisol was higher on d 3 post-transport and reached its lowest concentration at the end of the growth phase period (d 210). From d 7 onwards, the cortisol concentration was similar to that on d $-1$. There was a trend towards a lower cortisol concentration during unloading than that on d 14 ($p = 0.058$). The glucose was greater during unloading (d 0), with lower concentrations on d 7 and d 14 post-transport. The total plasma proteins increased ($p < 0.01$) from d 0 to d 7 post-transport, and were lower ($p < 0.01$) in the afternoon blood samples (d 42 to d 210) than in the morning samples (d $-42$, d $-1$, d 3, d 7, and d 14). The plasma albumin concentration increased ($p < 0.01$) with the diet supplementation of the weaned calves (d $-42$ to d $-1$). At the same time, it responded negatively to the stress of road transport. There was a trend towards a lower albumin concentration ($p = 0.08$) on d 3 compared to d $-1$ (pre-transport), whereas it was lower ($p < 0.01$) on d 14. Moreover, the albumin concentration was lower ($p < 0.01$) in the afternoon (d 42 and d 210) than in the morning samples.

**Table 4.** Plasma concentrations of metabolites in young Nellore bulls subjected to different immunomodulatory feed supplementation strategies in the pre- and post-transport during the growth phase.

| Item | Treatments [1] | | | | SEM | *p*-Value [2] | | |
|---|---|---|---|---|---|---|---|---|
| | Con | Pre | Post | Growth | | T | P | T × P |
| Cortisol (ng/mL) | 66.0 | 71.8 | 73.1 | 67.8 | 7.2 | 0.85 | <0.01 | 0.93 |
| Albumin (g/L) | 19.3 | 19.2 | 18.6 | 19.5 | 0.37 | 0.30 | <0.01 | 0.85 |
| Total proteins (g/L) | 54.4 | 54.1 | 54,1 | 56.4 | 1 | 0.36 | <0.01 | 0.62 |
| Urea (mmol/L) | 4.73 | 4.42 | 4.73 | 4.93 | 0.19 | 0.39 | <0.01 | 0.88 |
| Creatinine (μmol/L) | 102.5 | 108 | 111 | 103.2 | 2 | 0.14 | <0.01 | 0.17 |
| Glucose (mg/dL) | 86.5 | 93.5 | 90.2 | 90.4 | 3.5 | 0.51 | <0.01 | 0.81 |
| Cholesterol (mmol/L) | 3.01 | 2.86 | 2.95 | 2.92 | 0.06 | 0.52 | <0.01 | 0.79 |
| AST (U/L) | 68.1 | 73.5 | 70.3 | 75.4 | 3.9 | 0.32 | <0.01 | 0.5 |
| Calcium (mmol/L) | 1.74 | 1.73 | 1.67 | 1.76 | 0.03 | 0.26 | <0.01 | 0.9 |
| Phosphorus (mmol/L) | 1.7 | 1.65 | 1.62 | 1.72 | 0.05 | 0.23 | <0.01 | 0.78 |
| Magnesium (mmol/L) | 0.912 | 0.918 | 0.887 | 0.924 | 0.03 | 0.71 | <0.01 | 0.55 |

AST = aspartate aminotransferase. [1] CON: no immunomodulatory supplementation; NUTRA pre: inclusion of NUTRA (10 g/100 kg BW/day) only in the pre-transport period for 42 days; NUTRA post: inclusion of NUTRA (10 g/100 kg BW/day) only in the post-transport period for 42 days; and NUTRA growth: inclusion of NUTRA (10 g/100 kg BW/day) during the whole experimental period (pre and post-transport). [2] T: treatment effect; P: sampling period effect; and T × P: effect of the treatment × period interaction. Note: blood samples were collected on days d $-42$, d $-1$, d 0, d 3, d 7, d 14, d 42, and d 210. Cortisol, albumin, total proteins, and urea were analyzed in all these periods. AST and glucose were analyzed on days d $-42$, d $-1$, d 0, d 3, d 7, d 14, and d 210. Creatinine, cholesterol, calcium, phosphorus, and magnesium were analyzed on days d $-42$, d $-1$, d 0, d 42, and d 210.

The plasma urea concentration increased ($p < 0.01$) during the experiment due to changes in the grazing and supplementation conditions. The urea concentration on d 0 (unloading) was lower than that on d $-1$ (one day before transport) and tended to be lower ($p = 0.06$) on d 0 than on d 14. The concentration of the enzyme aspartate aminotransferase was higher on d 0 (unloading; $p < 0.01$), with lower values on d $-42$, d $-1$, and d 210 ($p \leq 0.05$).

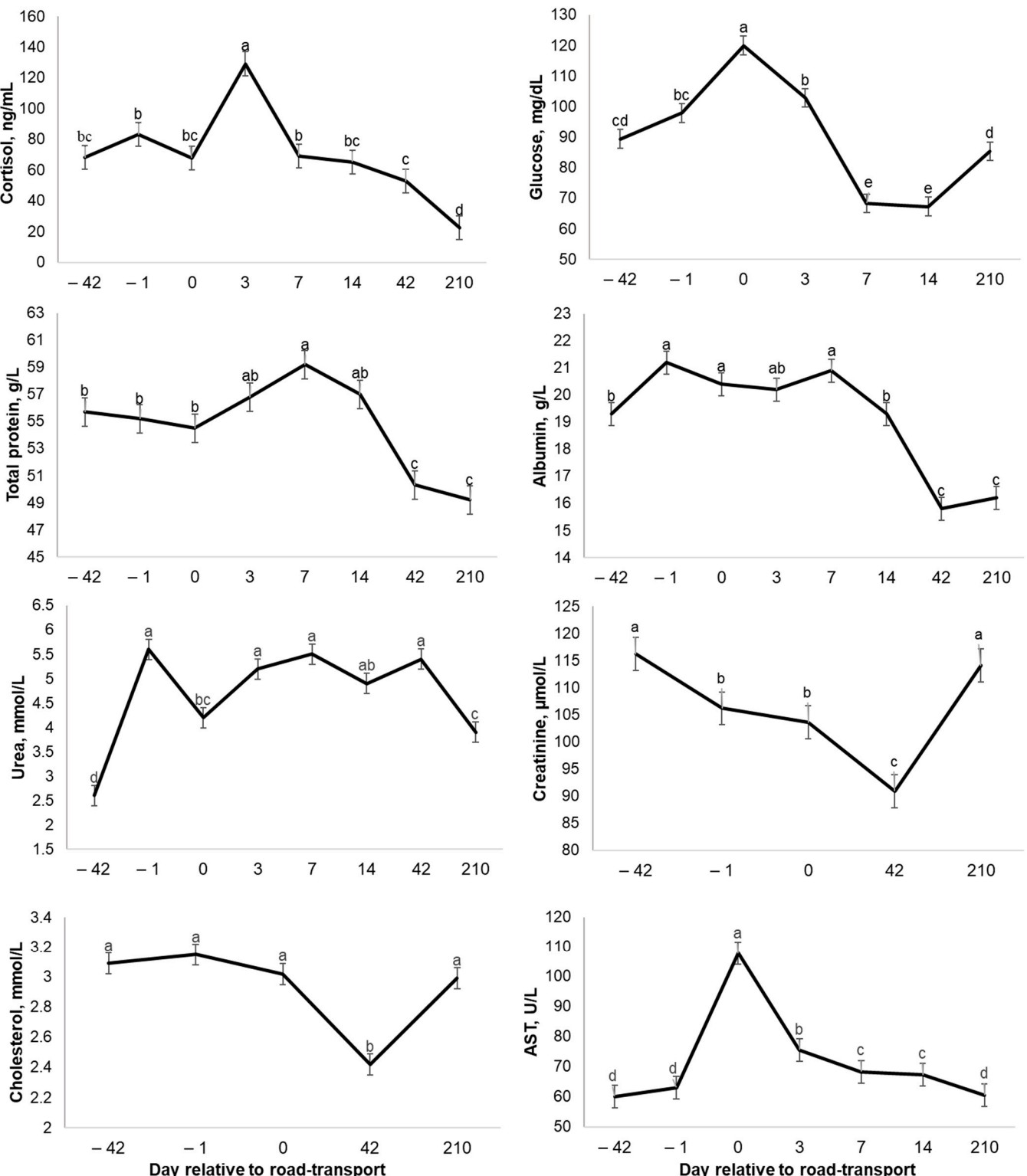

**Figure 3.** Plasma concentrations of cortisol, glucose, total proteins, albumin, urea, creatinine, and aspartate aminotransferase (AST) in young Nellore bulls subjected to different immunomodulatory feed supplementation strategies in the pre- and post-transport during the growth phase. Additive: NutraGen® added to the supplement at 10 g/100 kg BW/day. There was an effect of period on all variables ($p < 0.01$). [a,b,c,d,e] Values within the days followed by different letters differ at $p < 0.05$.

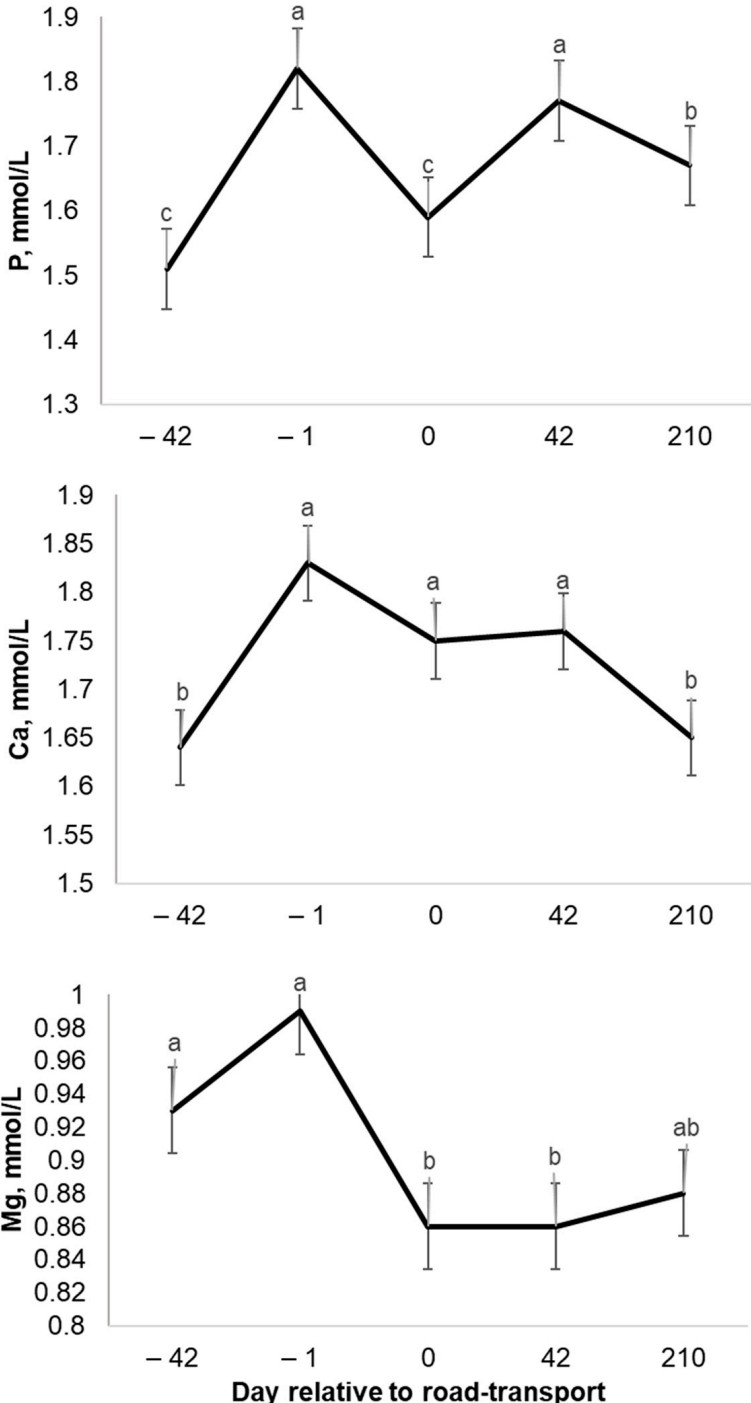

**Figure 4.** Plasma concentrations of phosphorus, calcium, and magnesium in young Nellore bulls subjected to different immunomodulatory feed supplementation strategies in the pre- and post-transport during the growth phase. Additive: NutraGen® added to the supplement at 10 g/100 kg BW/day. There was an effect of period on all variables ($p < 0.01$). [a,b,c] Values within the days followed by different letters differ at $p < 0.05$.

## 4. Discussion

This study shows the effects of NutraGen® supplementation on weaned Nellore cattle. In contrast to our hypothesis, we did not observe any additional benefits of NUTRA supplementation on the performances, metabolisms, and stress responses of Nellore calves that were transported in the post-weaning period. NUTRA supplementation benefits for animal performance were also not observed in Angus × Hereford calves that were purchased from



an auction market and transported for 12 h to a feedlot facility [11]. The authors also did not report any benefits of NUTRA on the feed efficiency, incidence, symptoms, and antibodies against bovine respiratory disease pathogens [11]. However, Lippolis et al. [11] started this NUTRA supplementation after the calves were exposed to immunological challenges caused by weaning, auction, transport, vaccination, mixing with animals from different sources, and feedlot entry [7]. In practical terms, a supplementation with NUTRA several weeks before a stress or immune challenge has been recommended [14].

In our study, for the NUTRA pre and NUTRA growth conditions, the animals received the feed additive supplementation for 42 days before being transported. These strategies allowed for the intake of NUTRA in advance of the transport (stressful event), although no effects on the animal performance and physiological parameters in plasma were observed. However, some particularities should be considered, such as the weaning method used (fence line weaning) and the period in which the calves were kept in the same paddock for weaning before being transported. These management practices are recommended to reduce stress, increasing the performance and health of beef calves compared to other methods of weaning and road transport immediately after weaning [15]. Moreover, the natural conditions of the grazing systems are usually less stressful for animals, which promotes an increased well-being compared to feedlot systems [25]. Studies in temperate regions with newly weaned animals upon feedlot entry report significant rates of morbidity and symptoms of bovine respiratory disease (BRD), resulting from the weakening of the immune system caused by stress [7]. It is worth mentioning that *Bos indicus* animals have a reduced acute-phase response to weaning and road transport stress compared to *Bos taurus* cattle, which indicates a greater resilience of Nellore cattle to stressful conditions [26]. According to Brandão et al. [27], NUTRA supplementation does not provide additional benefits for the innate immunocompetence of healthy beef cattle.

The lack of a treatment effect on the plasma parameters indicates that, under the experimental conditions of the present study, the NUTRA supplementation did not modulate the physiological responses of the cattle. However, the significant effects of the day and sampling intervals on the plasma-related variables indicate that the calves were exposed to stress associated with the transport procedures. It is noteworthy that the blood parameters should be interpreted based on similar sampling times (d $-42$, d $-1$, d 3, d 7, and d 14 at 800 h; d 42 and d 210 at 1500 h), given the circadian nature of cortisol secretion and its influence on metabolism [4].

The cortisol was higher on d 3 after the transport, unlike the expected peak during unloading. However, we believe that the cortisol concentration peaked during the transport, prior to the blood sampling, so that the rise in the cortisol during transport quickly returned to normal levels soon after the event [5,28]. The greater glucose concentration value during the unloading and on d 3 was probably because of the secretion of glucocorticoids such as cortisol, altering the glucose metabolism, consequently increasing its concentration in the blood [4,5]. This result corroborates with the hypothesis that the cortisol was higher before the sampling procedures during the unloading. It suggests that there was an increase in the cortisol during the transport, higher than that on d 3, and that this peak occurred before the blood collection.

The sharp increase in the cortisol levels elicited by the transport procedures and feedlot entry is accompanied by a transient response of acute-phase proteins in the cattle, particularly of haptoglobin and ceruloplasmin [7]. At the same time, this pattern correlates negatively with the DM intake and animal performance, leading to a subsequent reduction in immunocompetence. Therefore, this increases the BRD occurrence and morbidity of feedlot cattle [7]. Although we have not evaluated these acute-phase proteins, their transient responses can be deduced from the reduction in the plasma albumin concentration, which is negatively correlated with the acute-phase proteins [17,29]. The albumin concentration tended to decrease after the transport on d 3 and was lowest on d 14, even though there was an increase in the total protein concentration during the same period. Based on these responses, we can consider that, in case of the impossibility of dosing these acute-phase

proteins in the blood to identify an inflammatory response, the simultaneous dosage of the albumin–total protein pair may indicate an acute-phase inflammatory response triggered by cortisol, released as a result of the stress associated with the road transport. However, studies that also evaluate the acute-phase proteins during these procedures are necessary to validate this hypothesis.

Although there was evidence for the occurrence of acute stress in our study, the animals were supplemented on a pasture for 42 days after weaning and before transport, and were kept in a pre-established group. Preconditioning is a technique for reducing the stress of newly weaned animals that will be later transported [30]. After the transport, the animals were also kept in the same group and the same environment as before. On the other hand, management practices such as weaning followed by immediate transport, livestock auctions, the mixing of animals from different sources, extended periods of water and feed deprivation, vaccinations following these events, and feedlot entry are characterized as potentially more stressful situations for cattle. Therefore, these events are enough to deplete the body reserves to the point of impairing the biological functions and bringing an animal into a pre-pathological or pathological state, thus reducing the animal performance [5,7]. It is speculated, therefore, that there has not been sufficient stress in the studied management conditions to observe the benefits of NUTRA in Nellore animals, such as those obtained in experiments with dairy cows under conditions of thermal stress, transition periods, and sanitary challenges [12,13,31].

Aspartate aminotransferase is an indicator of muscle damage and may increase with trauma and muscle exercise, which can occur during transport [32]. In our study, there was probably muscle damage during the transport, since the AST had higher value during unloading. Contrary to what was expected, the plasma urea concentrations did not increase during the transportation, but were reduced relative to the pre-transport (d −1). This reduction was probably associated with a lower feed intake, and, consequently, the CP on the day of transport [33]. The animals were taken from the pasture to the corral at 06:00, and the transport and sampling procedures finished at approximately 17:00. Takemoto et al. [34] transported Holstein steers for 24 h and observed an increase in their plasma urea concentrations during unloading and up to 7 days after the transport, preserving similar feeding conditions. According to these authors, transport stimulates the deamination of amino acids and the use of carbon chains in energy metabolism, whereas nitrogen is metabolized as urea. Knowles et al. [35] also observed an increase in the plasma urea concentrations immediately after the transport, which suggests an increase in muscle protein breakdown. These results suggest that the challenge imposed on animals during transport is not enough to deplete the body reserves to the point of increasing the levels of urea in the blood via the deamination of amino acids from the muscle. It reinforces that the stress promoted by transport is not severe enough to cause a positive response associated with the use of an ingredient that modulates and stimulates the immune response of cattle under stressful conditions [6,31].

## 5. Conclusions

Under the conditions of this experiment, the supplementation with NutraGen® at 10 g/100 kg BW/day did not improve the physiological parameters and performances of grazing young Nellore bulls that were transported for 8 h. However, the short period in which the animals were subjected to transportation stress and the non-stressful grazing environment may have reduced the adverse effects of this stress on the animals, thus reducing the potential benefits of using the feed additive immunomodulator. New studies should be conducted based on more challenging situations to assess how the use of an immunomodulatory feed additive could benefit the performance of beef cattle.

**Author Contributions:** Conceptualization, L.H.C.B., C.A.d.A.B., G.R.S. and F.D.d.R.; methodology, L.H.C.B., I.M.O. and L.F.P.; validation, L.H.C.B., I.M.F. and L.F.P.; formal analysis, L.H.C.B. and I.M.O.; investigation, I.M.F., M.J.I.d.A., S.T.R.d.A., L.C.A. and L.F.P.; resources, G.R.S. and F.D.d.R.; data curation, L.H.C.B., G.R.S., F.D.d.R. and C.A.d.A.B.; writing—original draft preparation, L.H.C.B.; writing—review and editing, L.H.C.B., G.R.S., I.M.F., M.J.I.d.A. and S.T.R.d.A.; project administration, F.D.d.R., G.R.S., I.M.O. and L.F.P.; funding acquisition, F.D.d.R. and G.R.S. All authors have read and agreed to the published version of the manuscript.

**Funding:** This study was made possible by grants from the São Paulo Research Foundation (FAPESP) #2017/50339-5 (PDIP) and #2018/20176-0 (Fourth author), and National Council of Technological and Scientific Development (CNPq, Brazil #141577/2020-7). We also thank Phibro Animal Health Brazil for their financial support. The research was conducted as part of the first author's thesis.

**Institutional Review Board Statement:** This study was carried out in accordance with the ethical principle established by the Brazilian Council for the Control of Animal Experimentation and approved by the Ethics Committee for the Use of Animals of the São Paulo State University, College of Agricultural and Veterinary Sciences, São Paulo, Brazil (process no. 006088/19).

**Informed Consent Statement:** Not applicable.

**Data Availability Statement:** Not applicable.

**Conflicts of Interest:** One of the co-authors is associated with the manufacturer of the feed additive tested.

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
