# Peer review of "The Strategic Use of an Immunomodulatory Feed Additive in Supplements for Grazing Young Nellore Bulls Transported after Weaning: Performance, Physiological, and Stress Parameters"

_agriculture, doi:10.3390/agriculture13051027_

Round 1

Reviewer 1 Report

Reviewer comments

Title: Strategic use of an immunomodulatory feed additive in supplements for grazing young Nellore bulls transported after weaning: performance and physiological parameters

This publication summarizes the findings about evaluation four different feeding strategies using an immunomodulatory feed additive (NutraGen®) for newly weaned Nellore cattle before and after road transport, on physiological parameters and performance during the growing phase on pastures.

In the title I recommend adding an evaluation of stress parameters: Strategic use of an immunomodulatory feed additive in supplements for grazing young Nellore bulls transported after weaning: performance, physiological and stress parameters.

In  the abstract is missing comparison of cortisol value between monitored groups. In line 27-28 is only general evaluation from all monitored groups.

The introduction and Material and methods are well written.

The results, the figure 2 is not clear and well readable. I recommend re-formatting it into columns. In figure 3 is evaluated only one group and is missing comparison with others groups.

The discussion is well written.

The conclusion is too general. I recommend breaking down the last two sentences into specific comments how the stressful conditions of Nellore cattle transportation can be reduced after the addition of an immunomodulator feed additive.

Reviewer 2 Report

Dear authors,

The manuscript " Strategic use of an immunomodulatory feed additive in supplements for grazing young Nellore bulls transported after weaning: performance and physiological parameters " is a well-conducted study that takes into consideration many parameters in groups and the analysis performed is clearly explained and the results are properly reported and discussed. However, it requires revision before it can be considered for publication.

Abstract:

Don’t use the brand names in the abstract. It is more appropriate to write it in the M&M section.  I recommend adding the average body weight and age of the animals at the time of blocking. Add some exact studied parameter values and/or their p-values. The conclusion here could be more specific with X g/kg NUTRA supplementation in the diet….

Introduction:

The introduction is well-written. However, how was it prepared? How it is related to physiological parameters modulation and stress alleviation? Give more details on the product otherwise, it is too much of a commercial publication. Although “NutraGen®” has similar properties as Omnigen-AF but it is recommended to replace it with the term other immunomodulatory product … or more specifically Omnigen-AF. Line 44.

Material and Method:

Recheck the paddocks per block and animals per paddock (L 109-110). I have a few questions here; What was the routine for grazing? Why the supplement was included once a day? What about the water availability at the farm and around transportation? What was the feeding and watering strategy on the transportation day?

Statistics:

What is Normally, the significance is considered for p<0.05 and not p≤ 0.05 (please check it and correct it accordingly). I suggest that you add some exact p-values or p<0.05/p>0.05 (e.g.) for single parameters in the whole manuscript. 

 Results:

Pay attention to table 3. Some information discussed in the results is not present in the table e.g  Line 237

Conclusion:
Again, the conclusion can be more specific with the NUTRA inclusion level.

References:

Correct reference No 24. 

Round 2

Reviewer 2 Report

Dear Editor

After reading the authors' replies to the comments and recommendations, I approve this manuscript for publishing in your prestigious Journal.